# The Relative Merits of Observational and Experimental Research: Four Key Principles for Optimising Observational Research Designs

**DOI:** 10.3390/nu14214649

**Published:** 2022-11-03

**Authors:** Robert Hamlin

**Affiliations:** Department of Marketing, University of Otago, P.O. Box 56, Dunedin 9054, New Zealand; rob.hamlin@otago.ac.nz; Tel.: +64-3479-8161

**Keywords:** observational, research, experimental, design, risk, uncertainty, reliability

## Abstract

The main barrier to the publication of observational research is a perceived inferiority to randomised designs with regard to the reliability of their conclusions. This commentary addresses this issue and makes a set of recommendations. It analyses the issue of research reliability in detail and fully describes the three sources of research unreliability (certainty, risk and uncertainty). Two of these (certainty and uncertainty) are not adequately addressed in most research texts. It establishes that randomised designs are vulnerable as observation studies to these two sources of unreliability, and are therefore not automatically superior to observational research in all research situations. Two key principles for reducing research unreliability are taken from R.A. Fisher’s early work on agricultural research. These principles and their application are described in detail. The principles are then developed into four key principles that observational researchers should follow when they are designing observational research exercises in nutrition. It notes that there is an optimal sample size for any particular research exercise that should not be exceeded. It concludes that best practice in observational research is to replicate this optimal sized observational exercise multiple times in order to establish reliability and credibility.

## 1. Introduction

‘Does A cause B’? is one of the most common questions that is asked within nutrition research. Usually ‘A’ is a dietary pattern, and ‘B’ is a health, development or morbidity outcome [1]. In agricultural nutrition, the standard approach to such questions is to use a randomised experimental design [2]. These research tools were in fact developed within agricultural science in the Nineteen Twenties for exactly this purpose [3]. It remains extremely difficult to publish agricultural research that makes causality inferences without using such a design [4]. Other scientific disciplines have enthusiastically borrowed these experimental tools from agricultural science [5].

However, in human research, ethical or practical issues often make it impossible to use a randomised design to address such ‘does A cause B’ type questions [6]. As scientific and social imperatives require that these research questions still have to be addressed somehow, a variety of alternative approaches have been developed that are broadly grouped under the description of ‘observational research’ [7] (Observational research is confusingly defined in two ways within human research. In business research and some branches of psychology, observational research is defined as research where human behaviour is observed in a non-intrusive manner (e.g., watching shopper behaviour in a supermarket or eye tracking) as opposed to an intrusive approach such as a questionnaire [8]. In disciplines such as medicine and nutrition ‘observational research’ is defined as research in which the subjects’ allocation to a treatment condition is not randomised, and may not be under the control of the researcher [9]. In every other respect an observational study may follow recognised experimental procedures—the lack of randomisation is the key point of difference. This article addresses the second, medical/nutrition, form of observational research). Despite the absolute requirement to use these techniques in research environments which make randomisation a practical impossibility, researchers in human nutrition face the problem that observational approaches are often considered to be inferior to the ‘gold’ standard’ randomised experimental techniques [10,11]. The situation is aggravated by the association of observational research with the rather unfortunately termed ‘retrospective convenience sampling’ [12].

This negative assessment of observational research continues to dominate, despite reviews of the relevant literature that have indicated that research based upon observational and randomised controlled experiments have a comparable level of reliability/consistency of outcome [13,14,15].

This lack of clear cut advantage for randomisation in these reviews may well be due to the fact any ‘randomised’ sample where less than 100% of those selected to participate actually do participate is not randomised, as the willingness to participate may be linked to the phenomena being studied which can create a non-participation bias [16]. It is a fact that in any society that is not a fully totalitarian state 100% participation of a randomly selected sample is very rarely achievable [17]. In practice, participation rates in ‘random’ nutrition research samples may be well under 80%, but the use of such samples continues to be supported [18,19].

This credibility gap between randomised and observational studies is both a problem and potentially a barrier to the production and publication of otherwise useful observational research. It is summed up well by Gershon [15]:


*“Despite the potential for observational studies to yield important information, clinicians tend to be reluctant to apply the results of observational studies into clinical practice. Methods of observational studies tend to be difficult to understand, and there is a common misconception that the validity of a study is determined entirely by the choice of study design.”*
[15] (p. 860)

Closing up this credibility gap is thus a priority for observational researchers in a competitive publication arena where their research may be disadvantaged if their approach has a perceived lack of credibility. The gap may be closed by progress in two directions—(1) by increasing the relative credibility of observational research, and (2) by reducing the relative credibility of experimental research when applied to equivalent questions in equivalent situations.

The former approach is well summarised in the book by Rosenbaum [20] and many of the (9000+) published research articles that cite this work. The latter approach may appear at first to be both negative and destructive. It is nevertheless justified if randomised experimental techniques are perceived to have specific powers that they simply do not possess when applied to human nutritional research.

This commentary article adopts both approaches in order to assist those who seek to publish observational research studies, but not via statistics. It explains why the randomisation process does not confer experimental infallibility, but only an advantage that applies in certain situations. It demonstrates that via an over-focus on statistical risk it is perfectly possible to create a seemingly ‘low risk’ randomised experiment that is actually extremely unreliable with regard to its outcomes.

It concludes that consequently it is perfectly possible that a well-designed observational experimental design will comfortably outperform a poorly designed randomised experimental design with regard to an equivalent research objective. It concludes with a set of principles for researchers who are designing observational studies that will enable them to increase the actual and perceived reliability and value of their research.

## 2. Certainty, Risk and Uncertainty in Experimental and Observational Research

On 2 February 2002 in a press briefing, the then US Secretary of Defence, Donald Rumsfeldt, made the following statement:


*“… as we know, there are known knowns; there are things we know we know. We also know there are known unknowns; that is to say we know there are some things we do not know. But there are also unknown unknowns—the ones we don’t know we don’t know … it is the latter category that tends to be the difficult ones.”*
[21] (p. 1)

While it has often been parodied, e.g., Seely [22], this statement efficiently sums up the situation faced by all researchers when they are setting up a research exercise. Any researcher will be dealing with three specific groups of knowledge when they are in this situation, which can be summarised for this purpose as below (Table 1). It is critical that researchers fully understand these three groups and how they relate to each other within a human research environment.

### 2.1. What We Know (Group 1—Certainty)

While it is often treated as a certainty, Group 1 information is not actually so. Previous research results that may be used as Group 1 information are reported either qualitatively with no measure of the probability of it being right, or quantitively, via a statistically derived ‘*p*’ value (the chance of it being incorrect), which is always greater than zero [23] (The author is aware that the definition and use of *p* values is dispute, e.g., Sorkin et al. [24], and that a liberty is taken by describing and applying them to the discussion in this rather general manner, but the issue is too complex to be addressed here). Assuming that *p* = 0 for this pre-existing information does not usually cause serious issues with the design and outcomes of causal research as long as *p* is small enough, but this is not always so. Structural Equation Modelling (SEM) is one widely used instance where it can give rise to significant validity issues in research reporting [25]. The quote below is from an article specifically written to defend the validity of SEM as a tool of casual research:


*“As we explained in the last section, researchers do not derive causal relations from an SEM. Rather, the SEM represents and relies upon the causal assumptions of the researcher. These assumptions derive from the research design, prior studies, scientific knowledge, logical arguments, temporal priorities, and other evidence that the researcher can marshal in support of them. The credibility of the SEM depends on the credibility of the causal assumptions in each application.”*
[26] (p. 309)

Thus, an SEM model relies upon a covariance matrix dataset, which contains no causal information whatsoever, which is then combined with the ‘credible’ causal assumptions of the researcher—normally made ‘credible’ and supported by cited results from prior research. Bolen and Pearl acknowledge this credibility generation later on the same page of their article. When they put an assumption-based arrow on a covariance-based relationship in an SEM model, the researcher that is constructing it is assuming that *p* = 0 for that relationship. In fact, *p* is never zero, and is never reported as such by prior primary research. It may be a very low number, but even if it is a very low number, the accumulated risk of the entire model being wrong can become significant if the SEM model is large and many such assumptions are made within it.

In a recent article in ‘Nutrients’ [27] (Figure 6, p. 18) present an SEM with 78 unidirectional arrows. Leaving all other matters aside, what is the chance of this model being ‘right’ with regard to just the causal direction of all these 78 arrows? If one sanguinely assumes a *p* value of 0.01 for all 78 individual causal assumptions, and a similar level for *p* in the research itself, the probability of this model being ‘right’ can be calculated as 0.99^79^ = 4.5%. This very low level of probability is not a marginal outcome, and it is based upon a universally accepted probability calculation [28] and an authoritative account in support of SEM describing how SEM uses information with a high *p* value to establish causality [26]. It becomes even more alarming when one considers that once published, such research can then be used as a ‘credible’ secondary causal assumption input to further related SEM based primary research with its reliability/validity as Group 1 information ‘prior research’ readjusted up from 4% to 100%.

The conclusion is that ‘certainty’ in research is never actually so, and that consequently the more ‘certainty’ that a researcher includes in their theoretical development, the less ‘certain’ the platform from which they will launch their own research becomes. This is not an issue that is restricted to SEM based research—SEM just makes the process and its consequences manifest. The conclusion is that theoretical simplicity closely equates to theoretical and research reliability.

### 2.2. What We Know We Don’t Know (Group 2—Risk)

Identifying and acquiring specific information that we know we do not know is the basis of any contribution made by either experimental or observational causal research. These Group 2 relationships will thus be clearly defined by the researcher, and an enormous literature exists as to how such relationships may then be studied by either approach, and how the risk relating to the reliability of any conclusions may be quantified by statistics and expressed as a *p* value [29].

Typically Group 2 relationships will be few in number in any causal research exercise because a trade off exists between the number of variables that may be studied and the amount of data required to generate a statistically significant result with regard to any conclusions drawn [30,31,32]. The amount of data required usually increases exponentially, as does the number of potential interactions between the variables [30,31,32]. So, for example, a 4^2^ full factorial with six levels of each variable and 30 observations in each cell would require 480 observations to fully compare the relationships between 2 independent variables and one dependent variable. By contrast a 4^4^ full factorial would require 7680 observations to study the relationships between four independent variables and one dependent variable to the same standard.

This has led to the development of techniques that use less data to achieve the same level of statistical significance to express the risk related to multiple causal relationships [33,34]. Unsurprisingly these techniques, such as Conjoint Analysis, have proved to be extremely popular with researchers [35,36]. However, there is no ‘free lunch’, once again there is a trade-off. Conjoint Analysis, for example, is based upon a fractional factorial design [37]. The researcher specifies which relationships are of interest, and the programme removes parts of the full factorial array that are not relevant to those relationships [36]. As with any fractional factorial design, the researcher thus chooses to ignore these excluded relationships, within the fractional design, usually via the (credible) assumption that their main effects and interactions are not significant [38].

By doing so the researcher chooses to not know something that they do not know. These relationships are removed from the risk calculations relating to the variables that are of interest to the researcher. They and their effects on the research outcomes do not however disappear! They are transformed from visible Group 2 knowledge (risk) to invisible Group 3 knowledge (uncertainty). If the researcher’s assumptions are wrong and these excluded relationships are significant, then they have the potential to significantly distort the outcomes of the apparently authoritative analysis of the risk related to the visible Group 2 relationships that are eventually reported by the researcher. While techniques such as Conjoint Analysis that routinely rely upon highly fractionated fractional factorial designs are vulnerable in this regard [38], it is rarely acknowledged with regard to results that rely upon them. As with the SEM example above, the *p* value associated with the conclusion is routinely readjusted to zero on citation, and it thus graduates to the status of Group 1 knowledge (certainty).

### 2.3. What We Don’t Know We Don’t Know (Group 3—Uncertainty)

This category of knowledge, as Donald Rumsfeldt observed, is the one that creates most difficulty. It is also invariably the largest category of knowledge in any ‘living’ research environment, and it is at its most complex in human research environments. Its impact on data cannot be separated or quantified and thus must be treated as uncertainty rather than risk.

To illustrate this, take the situation where a researcher wishes to study the causal relationship between fructose intake and attention span for adolescents. The sample will be 480 adolescents aged between 12 and 16. For each adolescent, measures for fructose intake and attention span are to be established by the researcher.

The researcher may also presume that other factors than fructose intake will have an effect on attention span, and they may seek to capture and control for the impact of these ‘extraneous’ variables by a variety of methods such as high order factorials and ANOVA, conjoint analysis or linear mixed model designs. Whatever method is used, the capacity to include additional variables is always restricted by the amount of information relating to the impact of an independent variable set that can be extracted from any dataset, and the conclusions relating to them that can have a meaningful measure of risk attached to them via a *p* value.

Thus, in this case the researcher designs the research to capture the main effects of three other extraneous independent variables in addition to fructose intake: parental education, household income and the child’s gender. These relationships thus become Group 2 information.

This accounts for four variables that might well significantly impact upon the relationship between sugar intake and attention span, but it leaves many others uncontrolled for and unaccounted for within the research environment. These Group 3 uncertainty inputs (variables) may include, but are by no means restricted to, the diet of the household (which includes many individual aspects), the number of siblings in the household, the school that the adolescent attends and level of physical activity, etc. etc. These Group 3 uncertainty variables may be colinear with one or more of the Group 2 variables, they may be anticolinear with them, or they may be simply unconnected (random).

To take ‘school attended’ for example—If the sample are drawn from a small number of equivalent schools, one of which has a ‘crusading’ attitude to attention span, this Group 3 variable is likely to have a significant impact upon the dataset depending upon how it ends up distributed within its groups. If the effect is ‘random’ in its impact in relation to any one of the Group 2 variables, the effect of it will end up in the error term, increasing the possibility of a Type II error with regard to that Group 2 variable (as it might be with regard to gender if the school is coeducational). If the impact is collinear with any one of the Group 2 variables, then its effect will end up in the variation that is attached to that variable, thus increasing the possibility of a Type I error (as it certainly will if the crusading school is single sex).

The key issue here is that the researcher simply does not know about these Group 3 uncertainty variables and their effects. Their ignorance of them is either absolute, or it is qualified because they have been forced to exclude them from the analysis. A researcher will be very fortunate indeed if one or more of these Group 3 uncertainty variables within their chosen human research environment do not have the capacity to significantly impact upon their research results. This researcher for example had an experimental research exercise on olive oil intake destroyed by a completely unsuspected but very strong aversion to Spanish olive oil within the research population. The incorporation of Spanish origin into the packaging of one of the four branded products involved (treated as an extraneous variable with which the ‘Spanish effect’ was fully colinear) produced a massive main effect for package treatment, and substantial primary and secondary interactions with other Group 2 variables that rendered the dataset useless.

Group 3 uncertainty variables will always be present in any living environment. Because they are unknown and uncontrolled for, they are incorrigible via any statistical technique that might reduce them to risk. Consequently, the uncertainty that they generate has the capacity to affect the reliability of both experimental and observational studies to a significant degree. To illustrate, this the fructose and attention span causal example introduced above will be used. Table 2 shows how the Group 3 uncertainty variable (school attended) would affect a comparable experimental and observational study if its impact was significant.

Experiments are distinguished from observational studies by the capacity of the researcher to randomly allocate to treatment conditions that they control. Table 2 shows that randomisation may confer a significant advantage over non-randomly allocated observation in an equivalent causal research situation. However, Table 2 also shows that while experimentation may confer advantage over observation in comparable situations, it is a case of ‘may’, and not ‘will’. Randomisation does not confer infallibility, and this is because researcher knowledge and control only relates to Group 2 variables and the random allocation of subjects to them. Control does not extend to any Group 3 variable and is thus not absolute in any human research situation. The outcome is that significant uncertainty, unlike significant risk, cannot be eliminated by random allocation.

Therefore, it is perfectly possible to design an experiment that is less reliable than an observational exercise when investigating causal relationships. Because it cannot be eliminated, how the uncertainty that is generated by Group 3 variables is managed at the design phase of research is one aspect that can significantly impact upon the reliability of causal research that is conducted using either experimental or observational techniques. Perhaps more than any other, it is this aspect of agricultural research method, the management of uncertainty, and the generation of the ‘clean’ data by design that can minimise uncertainty, that has failed to transfer to human research disciplines.

## 3. Managing Risk and Uncertainty in Experimental and Observational Research—Fisher’s Principals

The development of modern, systematic experimental technique for living environments is usually associated with the publication of “The design and analysis of experiments’ and ‘Statistical methods for research workers’ by Sir Ronald Fisher [30,38,39]. Although Fisher’s work is most heavily recognised and cited in the role of risk reduction and the manipulation of Group 2 variables via random allocation between treatments, Fisher also was well aware of the potential impact of Group 3 variables and uncertainty on experimental reliability. In order to design ‘effective’ experimental research that dealt with the issue of Group 3 variables and uncertainty, Fisher proposed two ‘main’ principles:


*“… the problem of designing economical and effective field experiments is reduced to two main principles (i) the division of the experimental area into plots as small as possible …; (ii) the use of [experimental] arrangements which eliminate a maximum fraction of soil heterogeneity, and yet provide a valid estimate of residual errors.”*
[40] (p. 510)

The overall objective of Fisher’s principles is very simple. They aim to minimise the contribution of Group 3 variation to the mean square for error in the analysis of variance table, as the mean square for error forms the denominator of the fraction that is used to calculate the F ratio for significance for any Group 2 variable. The mean square for the variance of that Group 2 variable forms the denominator of the fraction. Therefore, reducing Group 3 variation increases Group 2 ‘F’ ratios and thus their significance in the ANOVA table as expressed by a ‘*p*’ value. Fisher’s principles achieve this by increasing sample homogeneity, which is in turn achieved by reducing sample size.

Fisher’s second principle for experimental design for theory testing is also closely aligned with the much older and more general principal of parsimony in scientific theory generation known ‘Occam’s Razor, which is usually stated as: *“Entities are not to be multiplied without necessity” (Non sunt multiplicanda entia sine necessitate)* [41] (p. 483). Occam’s Razor, like Fisher’s principles, is not a ‘hard’ rule, but a general principle to be considered when conducting scientific research [42].

This is as far as Fisher ever went with regard to these two ‘main’ principles for dealing with Group 3 variation and uncertainty. Exactly why they were not developed further in his writing is a mystery, but Fisher may have assumed that these principles were so obvious to his audience of primarily agricultural researchers that no further development was necessary, and that the orally transmitted experimental ‘method’ discussed earlier in this article would suffice to ensure that these two principles were applied consistently to any experimental research design.

The author’s personal experience is that Fisher’s assumptions were justified with regard to agricultural research, but not the medical, biological and social sciences to which his experimental techniques were later transferred without their accompanying method. To a certain degree this may be due to the fact that the application of Fisher’s principles for the reduction of experimental uncertainty are also easier to visualise and understand in their original agricultural context, and so they will be initially explained in that context here (Figure 1).

Figure 1a shows a living environment, in this case an agricultural research paddock. On first inspection it might appear to be flat and uniform, but it actually has significant non-uniformities within it with regard to soil, elevation, slope, sunlight and wind. The researcher either does not know about these non-uniformities (e.g., the old watercourse) or simply has to put up with them (slope, elevation and wind) in certain circumstances. These are all Group 3 variables in any research design. While Fisher used the term ‘soil heterogeneity as the input he wished to eliminate, he would have been more correct to use the term ‘environmental heterogeneity’.

In Figure 1b, a 3 × 4 fractionally replicated Latin Square experiment that is able to separate the main effects of three independent Group 2 variables, with the ability to detect the presence of non-additivity (interaction) between them has been set up (Youden & Hunter 1955). The experiment follows Fisher’s first principle in that the individual plots (samples) are as small as it is possible to make them without creating significant ‘edge effects’ [43]. It also follows Fisher’s second principle in that this form of fractionally replicated Latin Square is the most efficient design for dealing with this set of three Group 2 variables and simple non-additivity [5]. In Figure 1b the researcher has used the small size to avoid non-uniformity of sun and wind, and they have also fortuitously avoided any variations due to the river bed, if they were not aware of it.

In Figure 1c the researcher has breached Fisher’s first principle in that the plot sizes of the experiment have been increased beyond the minimum on the basis of ‘the bigger the sample the better’ philosophy that dominates most experimental and observational research design. This increase in plot size may reduce random measurement error, thus reducing the proportion of variance ending up in the error term and thus potentially increasing the F ratios for the Group 2 variables. However, the increase in accuracy will be subject to diminishing returns.

Furthermore, the design now includes all the variations in Group 3 variables in the environment. This may do one of two things. Firstly, variation generated by the Group 3 variables may simply increase apparent random variation, which will reduce the F ratio and induce a Type I error. Secondly, as is shown in this case, Group 3 variation may fortuitously create an apparently systematic variation via collinearity with a Group 2 variable. As the old water course is under all the ‘level I’ treatments for the third Group 2 independent variable, all the variations due to this Group 3 variable will become collinear with those of the third Group 2 independent variable. This will apparently increase the F ratio for that variable, and also simultaneously reduce that for the Youden & Hunter test for non-additivity of effects thereby creating a significant potential for a Type II error. (The Youden and Hunter test for non-additivity [44] estimates experimental error directly by comparing replications of some treatment conditions in the design. Non-additivity is then estimated via the residual variation in the ANOVA table. In this case, the three main design plots for Group 2 Variable 3, treatment level I are all in the watercourse, while the single replication at this level is on the bottom left corner of the design on the elevated slope. This replicated level I plot is likely to return a significantly different result than the three main plots, thus erroneously increasing the test’s estimate of overall error, and concomitantly erroneously reducing its estimate of non-additivity.)

In Figure 1d the researcher, who is only interested in three Group 2 main effects and the presence or not of interaction between them, has breached Fisher’s second principle by using a less efficient ‘overkill’ design for this specific purpose. They are using an 3 × 3 × 3 full factorial, but with the initial small plot size. This design has theoretically greater statistical power with regard to Group 2 variation, and also has the capacity to identify and quantify first, second and third order interactions between them—information that they do not need. The outcome of this is the same as breaching Fisher’s first principle, in that major variations in Group 3 variables are incorporated into the enlarged dataset that is required by this design. It is purely a matter of chance as to whether this Group 3 variation will compromise the result by increasing apparent random error, but this risk increases exponentially with increasing sample size. The randomisation of plots over the larger area makes a Type II error much less likely, but the chance of a Type I error is still significantly increased.

The design of an experiment that breached both of Fisher’s principles by using both the larger design and the larger plot size cannot be shown in Figure 1 as it would be too large, but the experiment’s dataset would inevitably incorporate even greater Group 3 variation than is shown in the figure, with predictably dire results for the reliability of any research analysis of the Group 2 variables.

It is important to note that that Fisher’s principles do not dictate that all experiments should be exceedingly small. Scale does endow greater reliability, but not as a simple matter of course. This scale must be achieved via replication of individual exercises that do conform to Fisher’s principles. ‘Internal ‘intra-study’ replication, where a small-sample experimental exercise is repeated multiple times to contribute to a single result does not breach Fisher’s principles, and it increases accuracy, power and observable reliability. It is thus standard agricultural research practice. Intra-study replications in agricultural research routinely occur on a very large scale [45], but it is rare to see it in human research disciplines [46,47]. The process is shown in Figure 1e, where the experiment from Figure 1a is replicated three times. With this design, variation in environment can be partitioned in the analysis of variance table as a sum of squares for replication. A large/significant figure in this category (likely in the scenario shown in Figure 1e) may cause the researcher to conduct further investigations as to the potential impact of Group 3 variables on the overall result.

Figure 1f shows a situation that arises in human rather than agricultural research, but places it into the same context as the other examples. In agricultural research, participation of the selected population is normally one hundred percent. In human research this is very rarely the case, and participation rates normally fall well below this level. Figure 1f shows a situation where only around 25% of the potentially available research population is participating as a sample.

Fractional participation rates increase the effective size of the sample proportionately (shown by the dotted lines) of the actual plots from which the sample would be drawn. The reported sample numbers would make this look like the situation in Figure 1b, but when it is shown laid out in Figure 1f, it can be seen that the actual situation is more analogous to Figure 1c, with a very large underlying research population that incorporates the same level of Group 3 variance as Figure 1c, but without the advantage of greater actual sample size, thereby magnifying the potential effect of Group 3 variables beyond that in Figure 1c. The outcome is an effective breach of Fisher’s first principle, and an increased chance that both Type I and Type II errors will occur.

Subject participation rate is therefore a crucial factor when assessing the potential impact of Group 3 variables on experimental research reliability. This derivative of Fisher’s first principle holds whether the experimental analysis of Group 2 variation is based upon a randomised sample or not.

Moving forward from these specific agricultural examples, the general application of Fisher’s principles with regard to the sample size used in any experiment can be visualised as in Figure 2.

As sample size increases, then ‘ceteris paribus’, the risk (R) of making a Type I or II error with regard to any Group 2 variable decreases geometrically, and is expressed via statistics in a precise and authoritative manner by ‘*p*’ value. As a consequence of this precision, this risk can be represented by a fine ‘hard’ solid line (R) in Figure 2.

By contrast, the uncertainty that is generated by the influence of Group 3 variables within the sample increases as the sample size itself increases. Unlike risk, it cannot be analysed, and no specific source or probability can be assigned to it—yet its increase in any living environment is inevitable as sample size increases. As it is fundamentally amorphous in nature it cannot be expressed as a ‘hard’ line, but is shown as a shaded area (U) in Figure 2.

The overall unreliability of research (T) is the sum of these two inputs. It is not expressed as a line in Figure 2, but as a shape that starts as a hard black line when the sample size is small and risk is the dominant input, and as a wide shaded area as sample size increases and uncertainty become the dominant input. The shape of the unreliability plot (T) is significant. As risk reduces geometrically, and uncertainty increases at least linearly with sample size, unreliability (T) takes the form of an arc, with a specific minimum point ‘O’ on the sample size axis where risk and uncertainty contribute equally to unreliability.

This indicates that there is a theoretical ‘optimal’ sample size at which unreliability is at its lowest, which is represented by a point (O) at the bottom of the arc (T). ‘O’, however, is not the optimal size of any experimental design. The point where sample size reaches point ‘O’, uncertainty is also the point at which uncertainty becomes the dominant contributor to overall experimental unreliability. However, as uncertainty is amorphous, the exact or even approximate location of ‘O’, and the sample size that corresponds to it, therefore cannot be reliably established by the researcher.

Given that ‘O’ cannot be reliably located, then the researcher must endeavour to stay well on the right side of it. It is clear from Figure 2 that, if there is a choice that is to be made between them, then it is better to favour risk over uncertainty, and to design an experiment that has specific risk contributing the maximum, and amorphous uncertainty the minimum, amount to its overall experimental unreliability for a given and acceptable value of *p*.

The logical reaction of any experimental designer to this conclusion is to ‘hug’ the risk line (R). This means that the minimum sample size that is required to achieve an acceptable not minimal level of experimental risk is selected, and further scale is achieved by replication of the entire exercise. This point is represented by the vertical dotted line ‘S1′ for *p* = 0.10 if the designer takes this to be the required level of risk for the experiment. If the designer reduces *p* to 0.05 and increases the sample accordingly, then they reduce the apparent risk, but they do not know with any certainty whether they are doing the same for overall unreliability, as uncertainty is now contributing more to the overall unreliability of the experiment (line S2). If risk is further reduced to *p* = 0.01, then the geometric increase in the sample size required increases the impact of Group 3 variable derived uncertainty to the point that it generates an apparently lower risk experiment that actually has a significantly higher (but amorphous and hidden) level of overall unreliability (represented by the double headed arrow on line S3).

It is this logical design reaction to the situation outlined in Figure 1 that is expressed by Fisher in his two principles. It should be noted that the required risk is the cardinal input. The acceptable level of risk must be established first, and this choice should be driven by the research objectives and not by the research design process. Fisher’s principles are then applied to minimise the contribution of uncertainty to experimental designs that are capable of achieving that level of risk.

## 4. Certainty, Risk, Uncertainty and the Relative Merits of Experimentation and Observational Research

All the foregoing remarks apply equally to randomised experimental research, and also to observational research that uses any form of organised comparison as the basis for their conclusions. Indeed, many observational research designs are classical experimental designs in all facets bar the randomisation of their treatment conditions.

In both cases poor design that does not address the potential contribution of Group 1 (certainty) and Group 3 (uncertainty) variation to their data can produce a highly unreliable research outcome that can nevertheless report a low level of risk. This outcome is made even more undesirable when this unreliable outcome is authoritatively presented as a low-risk result on the basis of a design and statistical analysis that focusses purely on the contribution of Group 2 (risk) variation to the data. The situation is further aggravated if the practice becomes widespread, and if there is a lack of routine testing of such unreliable results via either intra-study or inter study replication.

The answer to this problem is the application of method to reduce uncertainty and thus unreliability—Fisher’s two principles form only a small part of this body of method. At present the situation is that method is widely considered to be of little importance As Gershon et al. note [15] “*Methods of observational studies tend to be difficult to understand…”* Method is indeed difficult to report as it is both complex and case specific. My personal experience is that I have struggled to retain any methodological commentary in any article that I have published in the human research literature—It is just not perceived to be important by reviewers and editors—and thus presumably not worth understanding. Consequently, deletion is its routine fate.

One of the main barriers to the use, reporting and propagation of good method is that it is a fungible entity. While the techniques from Figure 1 such as Latin Square or ANOVA may applied to thousands of research exercises via a single, specific set of written rules, method is applied to research designs on a case-by-case basis via flexible and often unwritten guidelines. This is why ‘Fisher’s principles’, are principles and not rules. Thus, this article concludes by developing Fisher’s principles into a set of four methodological ‘principles’ for conducting observational research in nutrition—and for subsequently engaging with editors and reviewers:

**Principal 1.** 
*Randomisation confers advantage over observation in specific situations rather than absolute infallibility. Therefore a researcher may make a reasonable choice between them when designing an experiment to maximise reliability.*


Many observational studies are conducted because random allocation is not possible. If this is the case, then the use of observation may not need to be justified. If, however, the researcher faces the option of either a randomised or observational approach, then they need to look very carefully at whether the random design actually offers the prospect of a more reliable result. Ceteris paribus it does, but if randomisation is going to require a larger/less efficient design, or makes recruitment more difficult, thereby increasing the effective size of the individual samples, then level of uncertainty will be increased within the results t the degree that a reduction in reliability might reasonably be assumed. An observational approach may thus be justified via Fisher’s first or second principles.

**Principle 2.** 
*Theoretical simplicity confers reliability. Therefore simpler theories and designs should be favoured.*


All theoretical development involves an assumption of certainty for inputs when reality falls (slightly) short of this. This is not an issue when the inputs and assumptions related to the research theory are few, but can become an issue if a large number are involved.

There is no free lunch in science. The more hypotheses that the researcher seeks to test, the larger and more elaborate the research design and sample will have to be. Elaborate instruments make more assumptions and also tend to reduce participation, thus increasing effective individual sample size. All of these increase the level of uncertainty, and thus unreliability, for any research exercise.

The researcher should therefore use the simplest theory and related research design that is capable of addressing their specific research objectives.

**Principle 3.** 
*There is an optimal sample size for maximum reliability—Big is not always better. Therefore the minimum sample size necessary to achieve a determined level of risk for any individual exercise should be selected.*


The researcher should aim to use the smallest and most homogenous sample that is capable of delivering the required level of risk for a specific research design derived from Principle 2 above. Using a larger sample than is absolutely required inevitably decreases the level of homogeneity within the sample that can be achieved by the researcher, and thereby increases the uncertainty of Group 3 variables that are outside the control or awareness of the researcher. Unlike risk, uncertainty cannot be estimated, so the logical approach is not to increase sample size beyond the point at which risk is at the required level.

**Principle 4.** 
*Scale is achieved by intra-study replication—more is always better. Therefore, multiple replications should be the norm in observational research exercises.*


While there is an optimal sample size to an individual experimental/observational research exercise, the same does not apply to the research sample as a whole if scale is achieved by intra-study replication. Any observational exercise should be fully replicated at least once, and preferably multiple times within any study that is being prepared for publication. Replication can be captured within a statistical exercise and can thus be used to significantly reduce the estimate of risk related to Group 2 variables.

Far more importantly for observational researchers, replication stability also confers a subjective test of overall reliability of their research, and thus the potential uncertainty generated by Group 3 variables. A simple observational exercise that conforms with Principles 1–3 that is replicated three times with demonstrated stability to replication has a far more value, and thus a far higher chance of being published than a single more elaborate and ‘messy’ observational exercise that might occupy the same resource and dataset.

Clearly the research may not be stable to replication. However, this would be an important finding in and of itself, and the result may allow the researcher to develop some useful conclusions as to why this result occurred, what its implications are, and which Group 3 variable might be responsible for it. The work thus remains publishable. This is a better situation than that faced by the author of the single large and messy exercise noted above—The Group 3 variation would be undetected in their data. Consequently, the outcome would be an inconclusive/unpublishable result and potentially a Type 1 error.

## 5. Conclusions

Observational researchers will always have to face challenges with regard to the perceived reliability of their research. As they defend their work it is important for them to note that random designs are not infallible and that observational designs are therefore not necessarily less reliable than their randomised counterparts. Observation thus represents a logical path to reliability in many circumstances. If they follow the four principles above, then their work should have a demonstrably adequate level of reliability to survive these challenges and to make a contribution to the research literature.

Publishing experimental research of this type that takes a balanced approach to maximising experimental reliability by minimising both risk and uncertainty is likely to remain a challenging process in the immediate future. This is largely due to an unbalanced focus by reviewers, book authors and editors on statistical techniques that focus on the reduction of risk over any other source of experimental error [48].

Perhaps the key conclusion is that replication is an essential aspect of both randomised and observational research. The human research literature remains a highly hostile environment to inter-study replications of any type. Hopefully this will change. However, in the interim, intra-study replication faces no such barriers, and confers massive advantages, particularly to observational researchers. Some may approach replication with some trepidation. After forty years of commercial and academic research experience in both agricultural and human environments, my observation is that those who design replication based research exercises that conform to Fisher’s principles have much to gain and little to fear from it.

## 6. Final Thought: The Application of Fisher’s Principles to Recall Bias and within Individual Variation

One reviewer raised an important point with regard to the application of Fisher’s principles to two important nutritional variables:


*“There are some features on methods of data collection in nutritional studies that require attention, for example recall bias or within individual variation. The authors did not mention these at all.”*


The researcher operates in food marketing where both of these issues can cause major problems. There are significant differences between them. Recall bias as its name suggests is a systematic variation, where a reported phenomenon is consistently either magnified or reduced upon recollection within a sample. Bias of any type is a real issue when an absolute measure of a phenomenon is required (e.g., total sugar intake). However, due to its systematic nature, it would not necessarily be an issue if the research exercise involves a comparison between two closely comparable sample groups to measure the impact of an independent variable upon total sugar intake (e.g., an experiment/observational exercise where the impact of education on total sugar intake was studied by recruiting two groups with high and low education, and then asking them to report their sugar intake). If the two groups were comparable in their systematic recall bias, then the systematic recall effect would cancel out between the samples and would disappear in the analysis of the impact of education upon total sugar intake.

However, this requires that the two groups are truly comparable with regard to their bias. The chances of this occurring are increased in both random allocation (experimental) and systematic allocation (observational) environments if the sample sizes are kept as small as possible while all efforts are taken to achieve homogeneity within them. Response bias is a type 3 (uncertainty) variable. If the population from which the two samples above are drawn increases in size, then the two samples will inevitably become less homogenous in their characteristics. This also applies to their bias, which thus ceases to be homogenous response bias, and instead becomes increasingly random response variation—the impact of which, along with all the other type 3 uncertainty variables, now ends up in the error term of any analysis, thus decreasing the research reliability (See Figure 2). Response bias can thus best be managed using Fisher’s principles.

Similar comments can be made about within individual variation. The fact that people are not consistent in their behaviour is a massive issue in both nutrition and food marketing research. However, this seemingly random variation in behaviour is usually driven by distinct and predictable changes in behaviour which are driven by both time and circumstance/opportunity. For example, you consistently eat different food for breakfast and dinner (temporal pattern). You also consistently tend to eat more, and less responsibly, if you go out to eat (circumstance/opportunity pattern). If time/circumstance/opportunity for any group can be tightened up enough and made homogenous within that group, then this seemingly random within individual variation thus becomes a consistent within individual bias, and can be eliminated as a factor between study groups in the manner shown above.

Thus, within individual variation is a Group 3 (uncertainty) variable, and it too can be managed via Fisher’s principles. Although most research looks at recruiting demographically homogenous samples, less attention is paid to also recruiting samples that are also temporally and environmentally homogenous. Thus, a researcher should not only collect demographically homogenous samples but should also recruit temporally and environmentally homogenous samples by recruiting at the same time and location. This temporal and environmental uniformity has the effect of turning a significant proportion of within consumer variation into within consumer bias for any sample. The effect of this bias is then eliminated by the experimental/observational comparison. The small experiments/observational exercises are then replicated as many times as necessary to create the required sample size and Group 2 risk.

## Figures and Tables

**Figure 1 nutrients-14-04649-f001:**
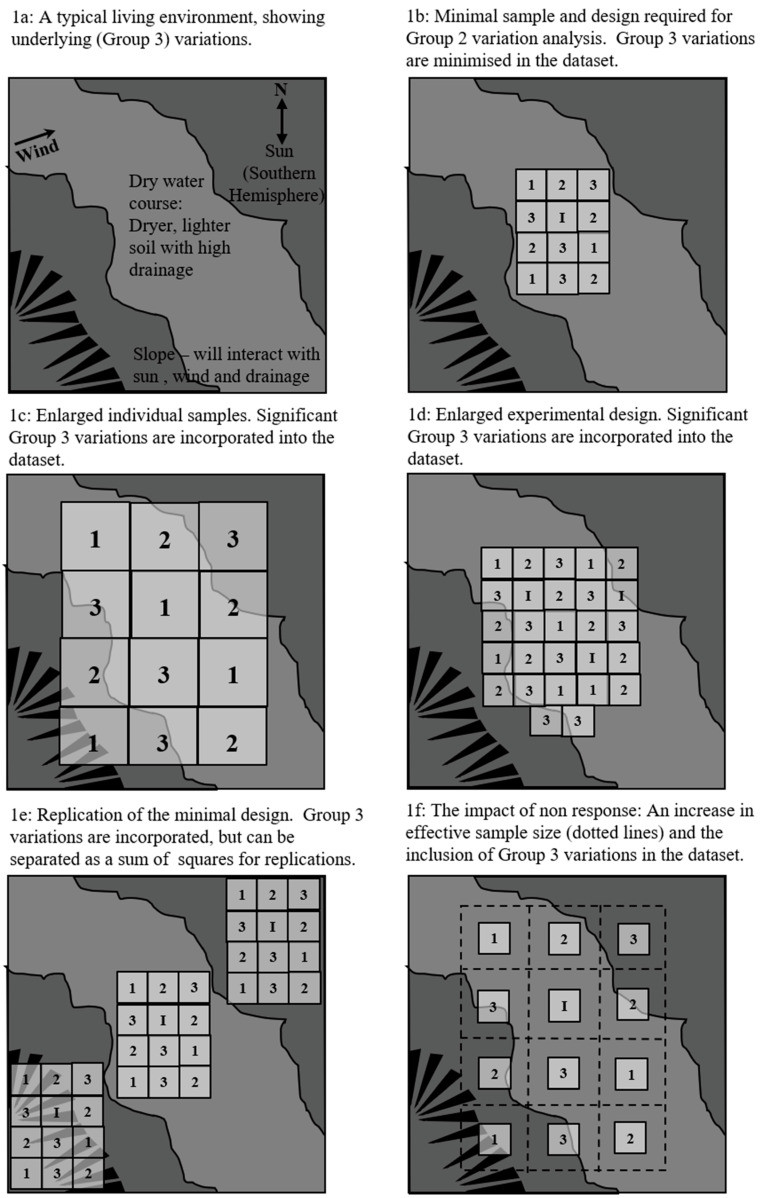
Fisher’s principles and Group 3 variables in the experimental environment.

**Figure 2 nutrients-14-04649-f002:**
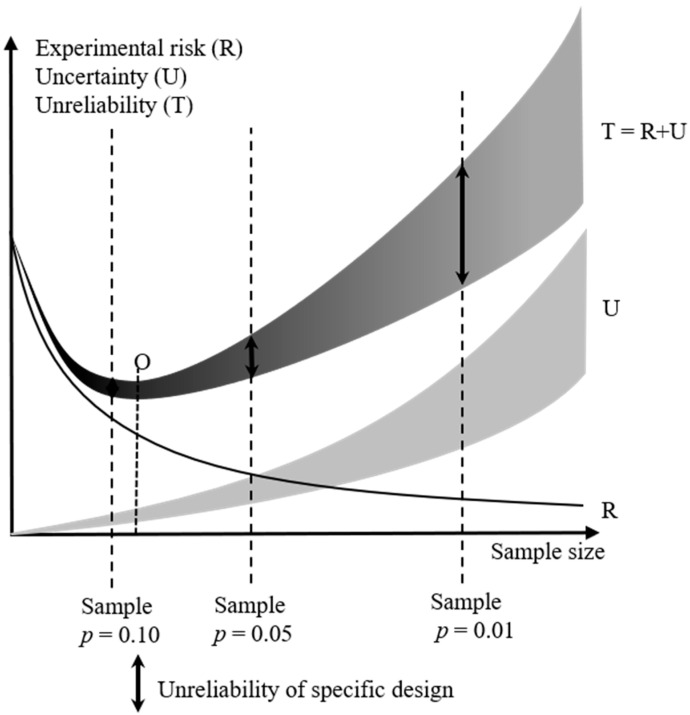
Graphical representation of the interaction of risk, uncertainty and unreliability as a function of experimental sample size.

**Table 1 nutrients-14-04649-t001:** The division of knowledge in research design (after Rumsfeldt).

Knowledge Group	Description	Definition
1. What we know “… *things we know we know …”*	The information available via earlier research and observation. Not actually a certainty (*p* = 0), but routinely treated as such.	Certainty
2. What we know we don’t know “… *we know there are some things we do not know …”*	The target relationship(s) of the research, and potentially a small number of other relationships and interactions. Usually quantified and described in the reporting process via a *p* value (*p* < *x*)	Risk
3. What we don’t know we don’t know *“… the ones we don’t know we don’t know …”*	All other relationships and interactions within the proposed dataset, including interactions of these unknown variables with the variables in Group 2 above. These cannot be specifically described or quantified. Additionally their potential impact is not usually discussed in any depth, or not at all, at any stage in the research design or reporting process.	Uncertainty

**Table 2 nutrients-14-04649-t002:** The impact of Group 3 uncertainty variables on experimental and observational research outcomes.

Experimental Study	Observational Study
***Design:*** 2^4^ factorial design—480 subjects recruited as eight matched groups of 60 on the basis of parental education, household income and gender. Within each group 30 randomly allocated to a high fructose diet and 30 to a low one, and attention span observed.	***Design:*** 480 subjects recruited as eight matched groups of 60 on the basis of parental education, household income and gender. Each group of 60 divided up into two groups of 30 (high and low) on the basis of their reported fructose consumption and attention span observed.
***Random impact Group 3 uncertainty input (school attended):*** The school attended effect will uniformly increase variation within the two randomly allocated experimental groups for high and low fructose diet. This increase in variation will end up in the error term of the analysis of variance, reducing the F ratio for fructose intake and for parental education, income and gender (trending to a Type I error). As the groups for parental education, household income and the child’s gender are not randomly allocated, the school effect will either end up in the error term of the analysis of variance thereby depressing the F ratio for parental education, income and gender if it is not colinear, or it will end up in the error that is related to these variables, and thus increase the F ratio if it is colinear. Therefore, results could trend towards a Type I or Type II error with regard to any or all of these Group 2 variables, depending on the level of and nature of the collinearity between them and the Group 3 variable. The school effect would be likely to be strongly colinear with all of these three Group 2 variables if the attention span crusading school was perceived to be the ‘good’ school in the area.	***Random impact Group 3 uncertainty input (school):*** The school attended variable will impact upon the parental education, household income and child’s gender variables exactly as it does in the experimental design opposite. The impact of the school attended variable upon the fructose intake variable will depend upon its degree of collinearity with it. If it is not collinear, then the allocation to the two groups will effectively be random, and the variation will thus end up in the error term depressing the F ratio for fructose intake, and tending towards a Type I error. If school attended has any collinearity with fructose intake, then the allocation will not be random and the impact of school attended will be apportioned into the variation associated with fructose intake. Depending whether the effect of school attended is complementary or anticomplementary to the effect of fructose intake, the result is a trend towards either a Type I (suppressed F ratio) or a Type II error (increased F ratio).

## Data Availability

Not applicable.

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
