# Peer review of "The Relative Merits of Observational and Experimental Research: Four Key Principles for Optimising Observational Research Designs"

_nutrients, 2022, doi:10.3390/nu14214649_

Round 1

Reviewer 1 Report

Congratulations to the author. I thoroughly enjoyed reading this paper. This is an important piece of work, not just for researchers in nutrition. 

As someone who just had a proposed clinical practice change thrown out the window because the organisational lead stated there were no RCT level of evidence supporting the practice change despite 1) it being unethical to test this practice experimentally, and 2) the a large amount of replicated observational research data summarised into systematic reviews and meta-analyses demonstrating significantly improved patient outcomes – I CANT THANK YOU ENOUGH! 

My comments are:

·      I am not sure that the title of the paper accurately describes the paper as there is a large portion of the paper discussing experimental research design

·      Sections 2 and 3, which make up the bulk of the paper, also specifically mention experimental research design in the section headings, albeit some text is about observational design within these sections. Section 4 starts saying all that has previously been said applies equally to experimental and observation research. Hence, would it be better to say something about observation research in section 2 and 3 headings - if the author wants the paper to be primarily about observational research, as per the current title of the manuscript? 

·      There are other pros and cons of experimental versus observational research other than their statistical strengths and flaws. For example, registries (nutrition and otherwise) are increasingly being used which provide already collected data that eases the data collection burden for observational studies in terms of human and time costs, that cannot be afforded to experimental studies . Would it be worth discussing these? However, I understand that the author may wish to keep the manuscript statistically pure.  

·      I learnt some new things, especially about uncertainty (thanks), however, there were some parts of this manuscript that were beyond my developing statistic skill set, I hope my other comments are worth considering.

Author Response

Dear Reviewer,

My responses are contained in the attached file.  I thank you for giving your time to review this article.

Reviewer 2 Report

I reviewed a paper entitled Publishing observational research:

Four key principles for research design.

I think it is a very interesting work, which offers important points of reflection with respect to the debate on the recognition of the reliability of non-randomized studies or between studies defined as experimental and observational studies.

I can only suggest always using the same expression for collinearity (sometimes it is expressed as colinearity) and a small note to correct "Fischer's" principles pag 10 line 308.

Author Response

Thank you for your comments both here and below.  They are greatly appreciated.  My responses are in red.

Reviewer 3 Report

This manuscript talks about presenting observational studies in nutrition/nutrients. But the manuscript seemed to discuss general issues instead of specifically on nutrition-related issues. There are some features on methods of data collection in nutritional studies that require attention, for example recall bias or within individual variation. Authors did not mention that at all. There have been a lot of epidemiological studies in nutrition. There were a couple of books on nutritional epidemiology, which provide solid methods in presenting and analyzing nutritional studies. This current manuscript seemed not aware of them.

Author Response

(The authors gave the same response as above.)

Round 2

Reviewer 3 Report

The author seemed familiar with food market research. Please limited the manuscript to that field. There is a well-known book on nutritional epidemiology by Walter Willett, which addresses issues on recall bias and within individual variation. The author did not know that.